# Predicting entrepreneur fundraising success from focus group EEG data

Jin Ho Yun[1,2*], Sohvi Heaton[3], JuiHsuan Sharon Wong[4], Peter Klein[3,5], Michael L. Platt[1,6,7,8]

1 Wharton Neuroscience Initiative, Wharton School of Business, University of Pennsylvania, Philadelphia, Pennsylvania, United States of America, 2 Department of Marketing, College of Business, New Mexico State University, Las Cruces, New Mexico, United States of America, 3 Department of Entrepreneurship and Corporate Innovation, Hankamer School of Business, Baylor University, Waco, Texas, United States of America, 4 Department of Biomedical Engineering, Weldon School of Biomedical Engineering, Purdue University, West Lafayette, Indiana, United States of America, 5 Department of Strategy and Management, Norwegian School of Economics, Bergen, Norway, 6 Department of Neuroscience, Perelman School of Medicine, University of Pennsylvania, Philadelphia, Pennsylvania, United States of America, 7 Department of Psychology, School of Arts and Sciences, University of Pennsylvania, Philadelphia, Pennsylvania, United States of America, 8 Marketing Department, Wharton School of Business, University of Pennsylvania, Philadelphia, Pennsylvania, United States of America

* drjyun@nmsu.edu

## Abstract

We examined how electroencephalographic (EEG) brain activity responded to entrepreneurial pitches for venture funding and whether EEG signals from a small laboratory focus group (N = 28) could be used to predict subsequent investment outcomes. Specifically, we tested whether EEG signals of frontal alpha asymmetry and correlated EEG activity across evaluators (i.e., neural similarity) were associated with investment interest expressed by a larger population group (N = 497), as well as with the likelihood that a pitch secured a deal and, if so, the size of the investment. We found that alpha asymmetry in the first 10 seconds of a pitch predicted population-level investment interest and investment amounts, above and beyond standard self-report and textual measures. Furthermore, correlated neural activity among viewers—a measure of communication effectiveness—ramped up over time during the pitch presentation and predicted deal outcomes. These findings suggest that EEG signals in evaluators viewing entrepreneurial pitches provide early indicators of aggregate investment interest and fundraising outcomes. Together, our findings illuminate a future roadmap for reverse engineering entrepreneurs' pitches using brain signals to enhance funding success.

## Introduction

To attract investors and other stakeholders to their ventures, entrepreneurs must pitch their ideas to potentially skeptical audiences. Pitching is challenging due to the novelty, complexity, and uncertainty often associated with new ventures. Even the

**Data availability statement:** Relevant data are available through the Open Science Framework: https://osf.io/da42v/?view_only=ad7ee-950c3844ab49285f45f83c7899c.

**Funding:** This study was supported by the Ministry of Education of the Republic of Korea and the National Research Foundation of Korea in the form of a grant awarded to JHY (NRF-2025S1A5C3A02009153) and New Mexico State University in the form of a salary for JHY. This study was also supported by the Wharton Behavioral Lab, Wharton Neuroscience Initiative, R01MH095894, R01MH108627, R37MH109728, R21AG073958, R01MH118203,866 R56MH122819, and R01NS123054, in the form of a grant awarded to MLP and the University of Pennsylvania. The specific roles of this author are articulated in the 'author contributions' section. The funders had no role in study design, data collection and analysis, decision to publish, or preparation of the manuscript.

**Competing interests:** The authors have declared that no competing interests exist.

most seasoned investors struggle to evaluate and assess the quality of the ideas they receive [1]. Researchers have proposed various decision models and factors to predict the likelihood a pitch will attract venture capital, including the entrepreneur's technical competence [2], the uncertainty associated with the idea [3], the investor's familiarity with the investment decision [4], learning on the part of venture capitalists (VCs) [5], and pitcher's passion, preparedness, and commitment [6]. Furthermore, conjunctive models [7] and actuarial models [8] have been proposed to incorporate decision cues utilized by VCs to predict the likelihood of attracting funding and thus the success of early-stage ventures.

However, existing studies measure perceptions of pitches and investor decision cues based on *subjective,* self-reported information obtained through survey ratings. An alternative is to examine directly physiological signals of investor reactions to model more effectively and forecast more accurately the likelihood that pitchers and investors will agree to transact [9]. For instance, an investor's physiology, measured by heart rate variability or a trader's morning testosterone level, offers useful information about their performance on trading floors and the profitability of their investments, respectively [10,11]. It is becoming increasingly evident that the brain also harbors *objective*, yet hidden, information that could be beneficial even for professional investors [12].

Moreover, analyzing physiological cues, rather than subjectively stated attitudes and behaviors, may lead to results that generalize more accurately to out-of-sample scenarios. Recent work on physiological signals known as "neuroforecasting" [12–16] has shown that a small set of neural focus groups (20–50 participants) has the potential to predict market behavior on a vastly larger scale. Building upon this work, here we aim to explore the use of neuroforecasting to develop a more comprehensive and objective understanding of the factors that predict pitch success. Open questions include: 1) Can brain signals from investors accurately predict the success of entrepreneurs' pitches on both an individual and aggregate level? 2) How does the predictability of traditional assessments compare with that made from brain signals? 3) At what point during the entrepreneur's pitch do investors exhibit the strongest neural similarity—a measure of communication effectiveness—predicting real-world deal outcomes? Answering these questions, and others, provides a foundation for more robust and informed reverse engineering of pitches to make them more appealing, engaging, and ultimately more effective at raising capital.

To answer these questions, we investigate how brain signals derived from a small sample of individuals can inform our understanding of the aggregated evaluation of entrepreneurs' pitches, ultimately influencing the success of early-stage ventures. Importantly, we investigate the use of electrical brain activity as a scalable forecasting tool capable of predicting aggregate outcomes. We draw on insights from recent multidisciplinary findings in neuroforecasting related to diverse contexts, including predicting movie ticket sales, crowdfunding, video engagement, and stock performance [12,14,17–19]. To our knowledge, no research has yet explored the neuroforecasting of entrepreneurial pitch performance success in new venture creation, in contrast to previous neuroforecasting work focusing on transactions involving more

familiar goods or services. Prior studies of entrepreneurs' pitching and VC decision-making typically relied on post hoc methods [20], which suffer from incomplete recall of the pitch itself [8] and implicit biases (e.g., racial or gender) in the decision-making of venture capitalists [21]. These biases are consequential, as they can obscure the moment-to-moment psychological processes that actually drive funding decisions and limit the extent to which self-reports can be used to forecast which pitches will succeed in new contexts.

Here the analysis of moment-to-moment electroencephalography (EEG) data allows us to investigate the dynamically changing content of funders' mental states and their instantaneous responses at specific points during a pitch, without interrupting them to ask questions. For instance, EEG signals in response to audiovisual stimuli can unpack dynamic changes in various emotional responses (e.g., happiness or sadness) over time [22]. Relatedly, our work identifies the specific moments that capture investors' interest and relate to deal outcomes in real-time, which neither traditional self-reports nor functional MRI (fMRI) data (i.e., slow changes in blood flow that unfold over seconds) can achieve [23]. Both survey instruments and fMRI signals are less amenable to capturing moment-to-moment responses during pitch presentations, making it difficult to pinpoint the precise timing that forecasts outcomes, which may be crucial for reverse engineering pitch design and delivery for maximum effectiveness.

To close these gaps, we adopt a scalable EEG that measures electrical signals on the scalp, which arise from the cortical surface of investors' brains during real-world pitch viewing. By analyzing behavioral and electrical signals, specifically frontal alpha asymmetry (FAA) and neural similarity collected from 28 participants, we predict investors' interest in a population-level group consisting of 497 participants. By also examining moment-to-moment EEG data, we further identify specific time windows during which FAA signals, as well as correlated neural similarity *among* investors—a neural marker of communication effectiveness indicating shared perception, attention, emotion, or meaning [17,24,25]—predict real-world deal outcomes.

EEG metrics, such as FAA, have shown promise as a forecasting tool by predicting population-level outcomes from data measured in small focus groups. FAA is computed from the difference in activity between the left and right frontal hemispheres of the brain and is known to be associated with approach motivation [26,27]. Prior EEG studies in neuro-forecasting validated that FAA predicts population-level TV viewership [28] and music popularity on Spotify [29]. Based on these foundations, we propose that when entrepreneurs' pitches are engaging, FAA will increase. Given previous studies that have documented the superior ability of brain data to predict aggregate outcomes, we hypothesize that higher FAA levels will not only predict individual investor interest but also broader, population-level interest, as well as real-world outcomes of investment amounts, over and beyond what traditional self-reported or textual measures of pitch transcripts can inform.

Importantly, we anticipate that the initial impression formed within the first 10 seconds of a pitch (e.g., when entrepreneurs introduce themselves and propose an equity deal in exchange for funding) might be particularly diagnostic of aggregate investor interest. Important work on thin-slice judgements in impression formation shows that people intuitively form stable evaluations from brief exposure, and those early evaluations anchor subsequent judgments, even as more information is revealed [30,31]. For instance, an eye-tracking and facial-coding study finds that early expressions of joy and surprise within the first ten seconds of a commercial predict more sustained attention and a lower likelihood of skipping [32]. Likewise, an fMRI study of short documentary clips shows that neural activity in the nucleus accumbens and anterior insula—regions associated with positive and negative affective responses, respectively—within the first four seconds of video onset predicts whether viewers will later choose to skip the video [14].

We further predict that the neural similarity metric *among* viewers will predict real-world deal outcomes. This metric is a neural marker of communication effectiveness reflecting shared perception, attention, emotion, or meaning evoked in multiple individual brains [17,24,33]. Building on extensive prior work, correlated neural similarity increases as a narrative unfolds because stories guide listeners into increasingly aligned mental states, enabling shared interpretation and anticipation [34–37]. In other words, effective storytelling progressively "locks" brains together, creating a common

neural trajectory that reflects successful communication. Applied to pitching, we hypothesize that as entrepreneurs articulate their problem, solution, and value proposition, stronger storytelling will increase neural similarity among evaluators. By contrast, pitches that are less cohesive or compelling will generate weaker correlation in investors' neural responses. Consistent with these predictions, prior work has shown that movie trailers with less complexity and fewer words evoke higher EEG similarity in the alpha band and subsequently yield greater ticket sales across the broader market [17].

Taken together, here we aim to test whether neural data from a small focus group can effectively predict self-reported group-level choices (i.e., the expressed investment interest in the proposed business idea) and real-world deal outcomes. At the individual level, we hypothesize that the ability to predict investments using FAA signals will persist even when controlling for individual characteristics, including levels of dynamic capabilities (DC) (i.e., the ability to sense opportunities, seize identified opportunities, and transform to adapt to evolving environments) [38]. More importantly, early-stage signals of alpha asymmetry will predict investor interest at the population level. FAA signals, averaged across the entire pitch, will even predict funding amounts raised above and beyond traditional survey-based metrics and sentiment analysis. Finally, we predict that neural similarity across individual brains will predict deal outcomes as storytelling crescendos in the pitch presentation.

## Materials and methods

### Sample and general procedure

Participants with backgrounds in industry practices and/or business degrees pre-registered for the study via the Wharton Behavioral Lab subject pool at the University of Pennsylvania. Twenty-eight participants (Mean age: 29.43 ± 10.85; 14 Females; Ethnicity: 9 Whites, 14 Asians, 1 Black, 1 Hispanic, and 3 Others) were recruited for the study. Our target sample size was guided by a prior EEG inter-subject correlation (ISC) approach using 30 participants to predict the success of 24 Spotify music songs [25]. Consistent with these, we recruited 28 participants. Recent methodological work further supports that ISC (in response to dynamic videos) reaches acceptable reliability ($r > .70$) with approximately 25 participants [39], placing our sample within the empirically recommended range.

We then ran Monte Carlo simulations (5,000 iterations; two-sided $\alpha = 0.05$) to estimate power for detecting video-level correlations between ISC and behavioral outcomes (14 pitch videos; N = 28, yielding 378 subject pairs). Estimated power was 6.1% for a small effect ($\rho = 0.10$; 95% CI [5.45%, 6.78%]), 17.0% for a medium effect ($\rho = 0.30$; 95% CI [16.00%, 18.09%]), and 37.6% for a large effect ($\rho = 0.50$; 95% CI [36.25%, 38.93%]), confirming limited sensitivity for small-to-moderate effects. Thus, our design should be viewed as exploratory and hypothesis-generating.

For the main EEG task, each participant watched fourteen Shark Tank pitch videos in a controlled laboratory room while their EEG activity was monitored. For each participant, the order of these fourteen pitch videos was randomly presented. We also note that one subject watched only 10 videos due to technical errors related to the stimuli presentation. The task was presented using PsychoPy software (https://www.psychopy.org/).

After watching each pitch video, participants rated six questionnaires regarding their investment interest (e.g., I would be interested in seeing more information about this venture) and their deal prediction (i.e., "Do you think this company will get a deal?"). We also note that none of the participants had previously viewed the specific pitch videos selected for this study, although a few participants were familiar with the Shark Tank TV show.

The EEG setup procedure lasted approximately 10 minutes, and the main task took about 35 minutes to complete. Upon completion, participants filled out a 5-minute survey consisting of 14 items measuring individual dynamic capabilities (i.e., evaluating one's sensing, seizing, and transforming capabilities) [36] and basic demographics (age, gender, race) before they were excused. The entire experiment lasted around 50 minutes, and participants received $25 as compensation for participating. All participants were native English speakers with no known hearing or vision impairments. Data collection commenced in February 2024 and was completed in March 2024.

All experimental procedures were approved by the University of Pennsylvania Institutional Review Board (IRB Protocol #832844). Written informed consent was obtained from all participants prior to their participation in the study, in accordance with the university's human subjects protection guidelines.

## Experiment stimuli

Videos from *Shark Tank* served as our primary stimuli. The entrepreneurial pitches featured on the show reflect real-world scenarios in which founders present their business ideas to potential investors. This context is not only familiar to many viewers but also highly engaging. We selected fourteen *Shark Tank* pitch videos aired between 2022 and 2024. The videos were similar in duration (M = 85.36 seconds, SD = 17.21), though they varied along several dimensions (see Table 1). Specifically, the selection spanned a range of product categories, including consumer-packaged goods (e.g., burgers, coffee, drinks, food, fries, sheets, snacks, soda, and spoons) and durable or high-involvement products (e.g., art, apps, kitchenware, mental health, and nutrition). Moreover, the selected stimuli included equal representation of male and female entrepreneurs (seven each) and reflected racial diversity, with four Black, six White, and four Asian pitchers. This demographic balance helps mitigate potential bias due to gender or race, offering a more comprehensive view of affective responses across a diverse set of individuals.

## Neural data acquisition

Our primary measure of brain signals was based on EEG recordings collected from recipients of watching Shark Tank pitch videos. The EEG recording data were collected via the B-Alert X24 Series Mobile EEG system developed by Advanced Brain Monitoring (ABM) operating at a rate of 1024 samples per second. The EEG system comprised 20 electrodes located according to the international 10–20 system (frontal brain regions: Fp1, Fp2, Fz, F3, F4, F7, F8; central regions: Cz, C3, C4, T3, T4; parietal regions: Pz, P3, P4; temporal regions: T5, T6; and occipital regions: O1, O2, and auxiliary channel for EOG to monitor eye blinks), along with linked reference channels located behind each ear on the mastoid bone. The EOG (electrooculogram) channels were placed on the left and right horizontal sides of the eyes.

After participants donned the wireless EEG cap, a washable conductive gel was applied with a syringe at each channel site. The conductive gel serves as a medium to reduce electrical impedance between the scalp and the electrodes, thereby improving signal quality. It facilitates more stable and accurate detection of the scalp's bioelectrical activity by

**Table 1. Pitching video stimuli used in the study.**

| Product Type | Gender | Race | Offer | Deal Outcome | Investor Interest Mean Scores (N = 497) |
|---|---|---|---|---|---|
| Art | Female | Black | No | No | 3.46 |
| App | Male | White | Yes | No | 3.91 |
| Burger | Male | White | No | No | 4.34 |
| Coffee | Female | Black | Yes | Yes | 3.98 |
| Drink | Female | Asian | Yes | Yes | 4.08 |
| Food | Male | Asian | Yes | Yes | 3.09 |
| Fry | Female | White | Yes | Yes | 4.98 |
| Kitchen | Male | Asian | Yes | Yes | 4.37 |
| Mental | Female | White | No | No | 3.62 |
| Nutrition | Male | Black | Yes | Yes | 3.86 |
| Sheets | Male | Black | No | No | 3.88 |
| Snack | Female | White | No | No | 3.67 |
| Soda | Female | White | Yes | No | 3.10 |
| Spoon | Male | Asian | Yes | Yes | 4.68 |

bridging microscopic air gaps and minimizing signal loss due to skin resistance. Further, prior to commencing the pitch viewing task, we verified the functionality of each EEG channel connection by ensuring that the electrical impedance levels were below 40 kΩ.

**Neural data processing and analyses.** We used the MNE-Python package [40] for preprocessing and analyzing all EEG data. To preprocess the EEG data, we organized the data according to the BIDS (Brain Imaging Data Structure) directory. Subsequently, we applied a bandpass filter to the EEG data ranging from 0.3 to 50 Hz. Further, we excluded ten non-EEG channels (e.g., AUX1, AUX2, SystemTimestamp) and incorporated video onset markers using annotations. The EEG electrode types were configured, designating the EOG channel assigned for monitoring eye blinks. Video onset markers were incorporated based on the event information of each subject. For each video (or epoch), we initially employed the RANSAC algorithm (using the Autoreject package) [41] to detect and correct any faulty channels.

Subsequently, we decomposed the epoched data into 15 independent components using independent component analysis (ICA), and we regressed out irrelevant components, particularly those related to eye movements (EOG channels), using a z-score threshold of 1.96. Eye movement-related components were detected via correlations with the EOG channel (threshold = 1.96), and on average between 1 and 4 ICA components were excluded per participant and video epoch. Then, the cleaned EEG data were baseline-corrected from –0.2 to 0 seconds. The preprocessed EEG signals were further transformed into power spectra of low- to high-frequency components (i.e., theta, alpha, beta, and gamma bands) using the Fast Fourier Transform (FFT) across all EEG channels.

## Measures

**Investment interest in the laboratory sample.** A key decision for participants was to self-report their interest in investing in the business idea presented in each video stimulus. Drawn from Shane et al. [42], investor interest was assessed using five items (α = 0.90), each rated on a 7-point scale. The items include: Q1 "Regardless of the nature of the product/service being pitched, the delivery of this pitch was 1=very poor to 7=excellent", Q2 "I would be interested in seeing more information about this venture", Q3 "Based on the information at hand, I would be interested in investing in this company", Q4 "This company represents a good investment opportunity for me", Q5 "I would expect higher financial returns from investing in this company than in other startup companies" on a 7-point scale (1 = strongly disagree; 7 = strongly agree).

**Investment interest in the out-of-sample.** We also gathered out-of-sample online survey data from the Amazon Turk (MTurk) Connect (N = 497) platform with participants aged 38.64 ± 11.08, including 217 females. During the study, each participant watched seven randomly assigned Shark Tank pitch videos out of fourteen available videos. In total, responses were collected for all fourteen videos, matching those of the laboratory participants. They were asked to self-report five ratings on investor interest after viewing each video, following the same procedure to that used in the laboratory experiment.

**Real-world outcome.** Our real-world outcome measure was a binary outcome of the actual Shark Tank pitching results: funded [1] or not funded (0). We also collected the actual amount of funding secured in each deal, which averaged $364,000. The specific amounts raised by each pitcher are listed in Table 1.

**Frontal alpha asymmetry.** We measure frontal alpha asymmetry (FAA). To ensure the accuracy of our FAA measurements, we implemented several rigorous procedures. First, FAA was computed from the alpha band (8–13 Hz) using power spectral density (PSD) estimates for every pitching video (averaged over the entire duration) and for each subject. We employed Welch's method to estimate the PSD of the alpha frequency band. The PSD values from the F4 channel (right frontal area) and F3 channel (left frontal area) were log-transformed and then subtracted, where positive values indicate relatively greater left than right frontal activity (e.g., greater approach motivation).

To also capture dynamic neural responses, we employed time-frequency representation (TFR) analysis using morlet wavelets. Given the dynamic nature of the video stimuli, TFR provides a more robust measure than static PSD, allowing for a finer-grained temporal analysis of FAA fluctuations over time. Specifically, we used a 5-second moving window

for every second (or rolling window) to analyze the temporal dynamic changes in FAA signals across different pitching videos (i.e., 0–5 seconds, 1–6 seconds, …, and 58–63 seconds). Given that the duration of each pitch video ranges from 63 seconds to 112 seconds, we applied the rolling window approach up to the minimum length (i.e., 63 seconds) for the aggregate forecasting. We then derived the PSD reflecting the power at the alpha band frequency and applied a 5-second moving window across the EEG data.

**Correlated neural responses.** We also measured correlated neural responses (or neural similarity) across individuals using the inter-subject correlation (ISC) metric. This produces a group-level index of synchronized brains *across* individuals in response to each pitch video. EEG epochs were first downsampled to 250 Hz to reduce computational load and standardize sampling across participants. ISC was then computed as the average pairwise Pearson correlations across participants, resulting in a total of 378 comparisons (calculated from the half triangle of the full inter-subject matrix: $(28 - 27)/2 = 378$). This process allowed us to create pairwise correlations for every data point. Given the association between synchronized brains and attention to naturalistic visual stimuli, we specifically focused on mid-central EEG channels (Cz, C3, and C4) within the alpha band (8–13 Hz), following the same approach employed by Barnett and Cerf [15]. Here, in lieu of using a short-time fourier transform (STFT) of the raw EEG signal, we opted to apply TFR using Morlet wavelets. This choice was made due to the superior adaptability of Morlet wavelets in capturing non-stationary signals over time [43], such as those elicited by naturalistic, video stimuli.

**Self-reported prediction.** During our video watching task, participants reported their stated prediction on each video: "Do you think this company will get a deal? (your prediction about other investors' choice)" on a binary choice (yes or no).

**Sentiment and emotion in pitch transcripts.** We derived textual sentiment features from transcripts of the Shark Tank pitch videos using two widely used lexicon-based approaches: the Vader sentiment analyzer and the NRCLex library for emotion analysis. The Vader tool provides a continuous valence index that captures positive–negative polarity, while NRCLex produces categorical affect scores across basic emotions (e.g., joy, fear, anger) related to Shark Tank pitch presentations. Thus, we selected these tools since our stimuli used are brief, 1–2 minute monologic pitches, which are emotionally better aligned with how founders script claims—such as eliciting excitement (joy), credibility (trust), or risk framing (fear). While these tools offer interpretable and efficient sentiment estimates, we acknowledge that lexicon-based models may miss subtleties such as sarcasm, authenticity, or contextual nuance. Future work could incorporate richer linguistic features, such as LIWC categories or transformer-based models (e.g., BERT), to better capture the complexity of pitch language and its emotional framing.

We then followed a series of procedures. First, we initialized the sentiment intensity analyzer (Vader) and applied it to each script to calculate sentiment scores, distinguishing between *positive* and *negative* sentiments. Second, we used the NRCLex library to analyze the scripts and extract emotion scores, focusing on *fear, joy, sadness, anger, surprise, and trust*. By isolating these specific emotions, we facilitated a more streamlined analysis. This approach allowed us to quantify both the overall sentiment and specific emotional content within the pitch transcripts, providing valuable insights into the emotional and sentiment dynamics of Shark Tank pitches. Third, drawing from previous research suggesting that successful pitchers tend to use more words than unsuccessful ones [44], we calculated the total word count in each transcript using the linguistic inquiry and word count (LIWC) tool.

**Control variables.** We include the individuals' dynamic capabilities (DC) as a control variable because prior work suggests that individuals with higher dynamic capabilities might be more attuned to changes in the environment and more effective at capturing emerging opportunities [36]. In measuring DCs, we utilized a scale developed by Heaton and Teece [36], comprising fourteen 7-point Likert scales that assess sensing, seizing, and transforming capabilities. Example items include "I am quick to identify environmental changes" (sensing), "I am adept at reallocating and adapting resources when necessary" (seizing), and "I persistently advance change efforts, even in the presence of unforeseen disruptions" (transforming) ($\alpha = 0.88$). Participants rated their agreement with each statement on scale ranging from 1 (strongly disagree) to 5 (strongly agree).

## Empirical specification

We utilized the FAA metric to examine its potential in predicting investment interest within a laboratory sample. This was compared to behavioral measures such as individual traits of DC ratings and textual sentiment scores, using both power spectral density (PSD) averages and time-frequency representation (TFR) with rolling windows. Subsequently, we tested whether group-level FAA activity, derived from TFR rolling windows, could be extrapolated to predict out-of-sample investor interest (N = 497). In addition, we assessed the group-level FAA, averaged across the entire video, as a predictor of aggregate funding amounts, comparing its performance to stated predictions and textual sentiment scores. Finally, we investigated the potential of a supplementary group-level measure, synchronized brains, using TFR rolling windows to predict deal outcomes and funding amounts. Relevant data and scripts analyzed are available through the Open Science Framework: https://osf.io/da42v/.

## Results

### Individual prediction of investment interest

We utilized hierarchical linear models (or linear mixed effects models) with random intercepts for participants and pitch videos to compare the impact of three fixed predictors on investment interest in entrepreneurs' startups: [1] neural activity (i.e., frontal alpha asymmetry) during the entire duration of the pitch video), [2] sentiment and/or emotion scores of pitch transcripts (e.g., joy, surprise, trust, positive, negative), and [3] self-reported ratings on DC traits. First, we found that FAA signals alone throughout each video significantly predicted investor interest ($\beta = 0.331$, $t = 2.15$, $p = 0.032$). Even after controlling for joy sentiment scores ($\beta = -1.365$, $t = -0.34$, $p = 0.738$) and self-reported DC traits ($\beta = 0.422$, $t = 1.83$, $p = 0.079$), only FAA signals remained significant ($\beta = 0.326$, $t = 2.11$, $p = 0.035$). This model showed a marginal improvement over the model that included only DC ratings ($\chi^2$ [2] = 4.65, $p = 0.098$) or both DC ratings and joy sentiment scores ($\chi^2$ [1] = 4.53, $p = 0.033$) (see Table 2).

Second, even after controlling for age ($p = 0.138$) and gender ($p = 0.902$), FAA neural activity still remained significant ($\beta = 0.338$, $t = 2.20$, $p = 0.029$). Next, we analyzed the diversity in both participants' and entrepreneurs' race as potential effects on investor interest [45]. By introducing the interaction between participants' race and pitchers' race as a fixed-effect predictor, we observed that race did not have a significant impact (all $p$ values > 0.05), and the FAA predictions remained significant ($\beta = 0.375$, $t = 2.40$, $p = 0.017$).

Third, instead of using power spectral signals averaged across the video duration, we employed a TFR rolling window approach for every second (i.e., using a 5-second window size where each data point presents neural activity over

**Table 2. Individual prediction of investment interest.**

|  | (1) | (2) | (3) | (4) |
|---|---|---|---|---|
| Constant | 2.714 (0.236)** | 4.446 (0.546)*** | 4.324 (0.212)*** | 2.984 (0.987)** |
| Individuals' Dynamic Capability | 0.438 (0.236) |  |  | 0.422 (0.230) |
| Joy sentiment in transcript |  | −1.31 (4.00) |  | −1.365 (3.984) |
| Frontal alpha asymmetry |  |  | **0.331 (0.154)*** | **0.326 (0.154)*** |
| *AIC* | 1230.62 | 1228.23 | 1230.25 | 1227.40 |
| *Adjusted R²* | 0.460 | 0.461 | 0.450 | 0.465 |

*Notes:* The statistics presented are standardized coefficients and standard errors (SEs).
Significance levels are denoted as follows: *$p < 0.05$; **$p < 0.01$; ***$p < 0.001$.

consecutive intervals: the initial data point corresponds to neural activity from 0 to 5 seconds, the subsequent point over 1–6 seconds, and so forth). Using hierarchical linear models with random intercepts for participants and pitch videos, FAA signals observed during the rolling window within the first 4–8 seconds significantly predicted investor interest (β = 9.542, t = 2.48, p = 0.030), even after controlling for DC ratings (β = 0.438, t = 1.86, p = 0.074) and joy sentiment (β = 4.412, t = 1.09, p = 0.301). This initial 4–8-second period emerged as important, as it is when thin-slice impressions might be formed and when entrepreneurs' physical appearance, vocal tone, and proposed deal (i.e., the equity offered in exchange for funding) are first introduced in the pitch. Furthermore, this model indicated that FAA signals continued to enhance the explanatory power of the model, beyond self-reported assessments and textual sentiment measures. This led to a decrease in Akaike Information Criterion (AIC) to 1220.11 and an increase in predictive variance by 46.8%. Thus, greater FAA signals captured in early impressions to persuasive pitches predicted investment interest.

## Population-level prediction of investment interest

Considering that FAA signals can fluctuate throughout the course of a persuasive message, we investigate the point at which brain responses predict investor interest, leveraging fast temporal precision of EEG signals. That is, we can identify *when* predictive neural activity peaks, enabling us to predict out-of-sample measures.

Here we again computed the group-level FAA activity for every second using 5-second rolling windows (where the first data point indicates neural activity for 0–5 seconds, the second data point for 1–6 seconds, and so on) at an aggregate level. Subsequently, we computed the correlations between moment-to-moment FAA signals and investor interest (from population-level MTurk N = 497) to identify the temporal peaks of predicting investment interest.

While each pitch has its own unique style, affective FAA activity collected in the laboratory sample significantly predicted out-of-sample (N = 497) investment interests (r > 0.50, p < 0.05) within the first 4–8 seconds of a video pitch (see Fig 1). To further assess the robustness of this effect, we conducted nonparametric bootstrapping with 5,000 iterations. FAA correlations during 4–5s remained consistently positive (Window 4: r = 0.59, 95% CI [0.05, 0.85]; Window 5: r = 0.57, 95% CI [0.03, 0.84]). Hence, our TFR rolling window approach robustly demonstrated predictive power at both the individual and aggregate levels in predicting investment interest during the peak of the first impression. We also note that none

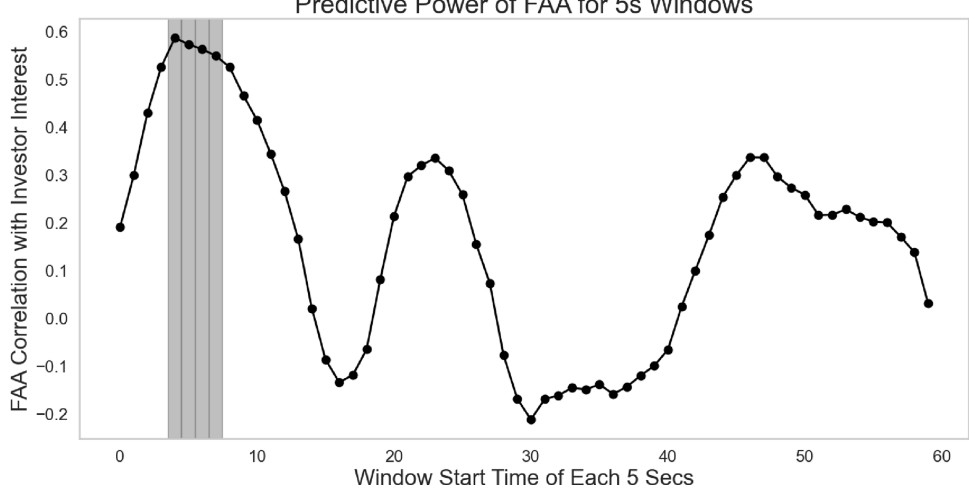

**Fig 1. Moment-to-moment FAA activity.** The shaded area represents the significance level for predicting population-level investment interest. FAA refers to frontal alpha asymmetry.

of the textual sentiments or emotions extracted from the analyzed pitch transcripts (e.g., joy, trust, and positive emotion) were significantly related to out-of-sample investor interest and deal outcomes (all $p$ values > 0.10).

## Aggregate forecasting of real-world outcomes

We next investigated whether our in-sample viewers' responses (neural and behavioral) were in any way linked to the eventual real-world outcomes. We explored whether FAA signals *averaged* over the entire video duration can forecast real-world fund amounts raised. We aimed to determine the predictive power of neural data above and beyond self-reported prediction (get a deal or not) and textual sentiments, thereby strengthening the robustness of our results on aggregate forecasting. We found that the average FAA activity (but not correlated brain signals: $r = 0.13$, $p > 0.10$) was the strongest predictor of aggregate fund amounts ($r = 0.66$, $p = 0.010$) compared to participants' stated prediction ($r = 0.46$, $p = 0.099$), and other textual sentiment or emotion scores extracted from the pitch transcripts (joy: $r = 0.41$, trust: $r = 0.38$, positive: $r = 0.04$; all $p$ values > 0.10). Importantly, average FAA activity forecasted fund amounts ($\beta = 1989.98$, $t = 3.19$, $p = 0.010$) over and beyond self-reported participants' prediction ($\beta = 417.28$, $t = 2.18$, $p = 0.054$) and joy sentiment ($\beta = 1898.42$, $t = 2.04$, $p = 0.069$) (see Table 3). Thus we find that *overall* FAA signals predicted aggregate fund amounts raised.

Despite these overall effects, when we employed the TFR rolling window analysis technique, none of the time windows of moment-to-moment FAA activity forecasted binary deal outcomes (funded or not funded; all $p$ values > 0.10). However, as shown in Fig 2, neural similarity (i.e., inter-subject correlation) is consistently higher for funded than unfunded pitches, and the two trajectories diverge markedly in the later portion of the pitch, particularly after the 30-second window. Fig 2 illustrates how engaging storytelling can gradually build neural similarity over time and increase the likelihood of securing a deal, consistent with neural similarity providing a biomarker of effective communication. Importantly, neural similarity significantly forecasted real-world deal outcomes around 47–50 seconds into pitches ($r > 0.50$, $p < 0.05$) (see Fig 3). Here we also ran nonparametric bootstrapping with 5,000 iterations: ISC correlations during 47–49s remained consistently positive (Window 47: $r = 0.59$, 95% CI [0.11, 0.87]; Window 48: $r = 0.62$, 95% CI [0.16, 0.87]; Window 49: $r = 0.62$, 95% CI [0.16, 0.87]).

Importantly, this similarity metric (measured using a 3-second rolling window: 0–3 seconds, 1–4 seconds, and so on) also marginally forecasted the actual funding amounts (averaging $364,000) at the 48-second time window ($r = 0.458$, $p < 0.10$). Economically, this translates into a 1.4% increase in the probability of securing a deal for each one standard

**Table 3. Forecasting aggregate funding amounts.**

|  | (1) | (2) | (3) | (4) |
|---|---|---|---|---|
| Constant | −55.753 (144.059) | −93.284 (183.737) | 476.245 (106.119)** | −4.362 (186.729) |
| Self-reported prediction | 498.638 (279.092) |  |  | 417.283 (191.245) |
| Joy sentiment in transcript |  | 2203.310 (1399.069) |  | 1898.417 (932.157) |
| Frontal alpha asymmetry |  |  | **2342.058 (760.625)*** | **1989.982 (624.024)*** |
| *AIC* | 199.70 | 191.40 | 185.90 | 181.20 |
| *Adjusted R²* | 0.144 | 0.101 | 0.395 | 0.608 |

*Notes:* The statistics presented are standardized coefficients and standard errors (SEs). Significance levels are denoted as follows: *$p < 0.05$; **$p < 0.01$; ***$p < 0.001$. Self-reported prediction refers to participants' prediction about whether deals are made or not. The FAA power signals were averaged across the whole video duration.

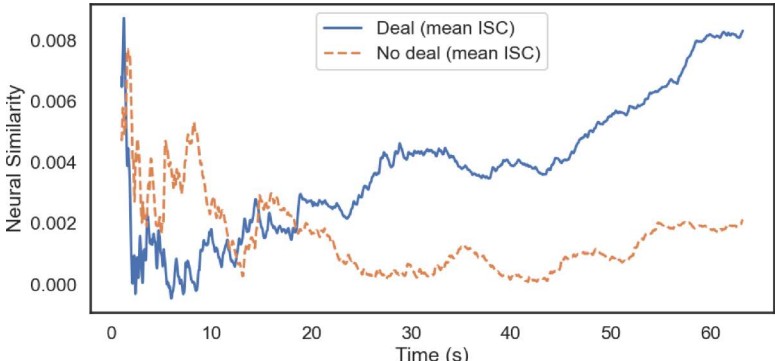

**Fig 2. Mean moment-to-moment neural similarity over time for funded vs. unfunded pitches.** ISC refers to inter-subject correlation (or neural similarity across viewers).

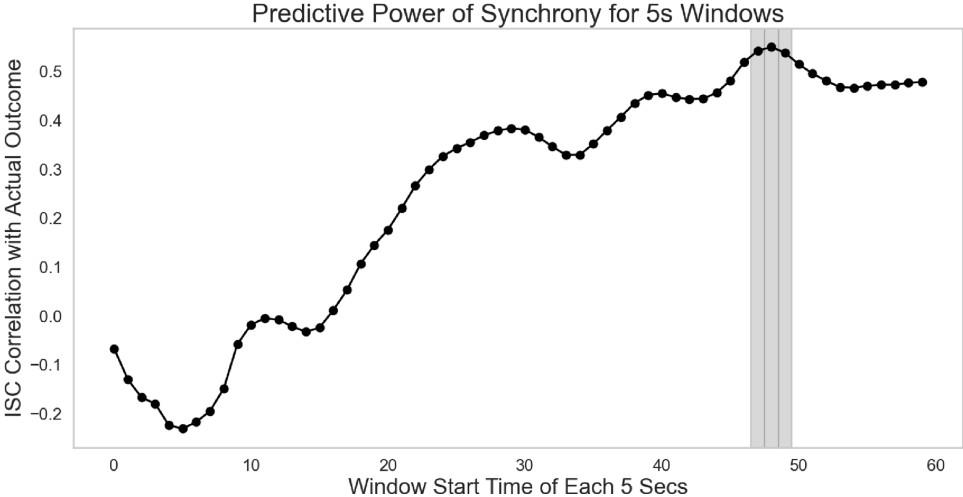

**Fig 3. Moment-to-moment correlated brain activity.** The shaded area represents the significance level for forecasting real-world deal outcomes. ISC refers to inter-subject correlation (or neural similarity across viewers).

deviation increase in neural similarity. We also explored whether FAA or neural similarity could forecast real-world outcomes for the two pitches (App and Soda) that were offered but declined. Our findings indicate that neither FAA nor neural similarity significantly forecasted deal outcomes. Nonetheless, neural similarity again peaked after 45 seconds ($r > 0.20$), although this increase was not statistically significant ($p > 0.10$). We also note that this metric did not predict out-of-sample investment interest within any of the time windows (all $p$ values $> 0.10$).

The correlation between real-world deal outcomes and neural similarity builds up over time as founders tell their stories and peaks when they reveal their product solutions. This collective response is a well-validated measure of communication effectiveness [34,37] that captures the overall persuasiveness of the founders' stories and the appeal of the product solution being presented—factors that individual FAA might miss. However, it is important to note that the real-world deal outcome on Shark Tank is often determined by one or two investors, which may not fully represent the aggregate decision. Nonetheless, this finding suggests that storytelling may be fundamental to persuading investors and that revelation of the product solution is a key moment for securing a deal.

Lastly, Fig 4 displays that the strongest neural similarity predicting both deal outcomes and investment amounts emerges over frontal–central regions, widely implicated in sustained attention, narrative integration, or evaluative processing. In naturalistic communication, correlated activity in these regions might reflect the extent to which listeners track a story in a similar way and converge on shared interpretations. These considerations suggest that the pitch narrative is engaging them in a common cognitive and emotional trajectory—an effect that translates into real-world funding decisions.

### Robustness checks

We also analyzed additional temporal dynamics of forecasting deal outcomes using *rank-ordered* correlation as well as 3-second rolling windows (0–3 seconds, 1–4 seconds, and so on) instead of 5-second windows employed in the main analysis. This robustness check is crucial, as shorter time intervals facilitate a more granular examination of the temporal dynamics at play, potentially capturing rapid fluctuations in message recipients' sentiment that could be overlooked with longer windows. The results remained robust: [1] using both rank-ordered correlations and 3-second rolling window showed that neural similarity across individuals accurately forecasted binary deal outcomes within 48–50 seconds ($r > 0.65$, $p < 0.05$) and [2] using rank-ordered correlations and 5-second rolling window also showed that this measure accurately forecasted deal outcomes within 48–50 and 59 seconds ($r > 0.59$, $p < 0.05$). FAA activity also marginally predicted out-of-sample interests within 4 seconds ($r = 0.48$, $p = 0.085$).

Finally, to test possible sensitivity to alpha band definitions, we reran the mixed effects model using FAA derived from 8–12 Hz instead of the canonical 8–13 Hz band. The results were almost identical: FAA in the 8–12 Hz range also predicted out-of-sample investor interest ($\beta = 0.332$, $t = 2.17$, $p = 0.031$), confirming that our FAA results are not driven by specific frequency cutoffs. Taken together, this consistency across different correlational metrics, time windows, and band definitions enhances the robustness of our findings.

### Discussion

Predicting the success of entrepreneurs' pitches is crucial for both entrepreneurs seeking funding and investors aiming to identify promising ventures. Evaluating a pitch is a complex and uncertain process that encompasses cognition, body language, and emotions, with responses being dynamic in nature. Our work has empirically shown that the scalable EEG metrics we tested offer predictive insights into population-level investment interests for the pitched business idea, as well as real-world outcomes.

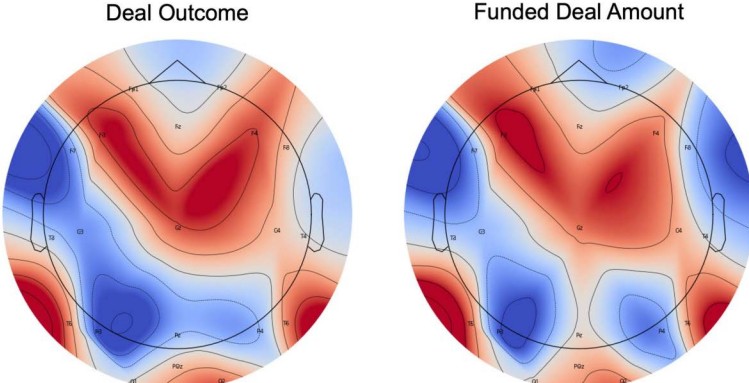

**Fig 4. Neural similarity predictive power across EEG electrodes.** 20 electrodes indicate the correlation values of correlated neural responses across individuals (averaged across the entire video duration) with both deal outcomes and funding amounts.

As anticipated, we found that neural data from a laboratory sample predicted aggregate outcomes in terms of expressed investment interest in the proposed business idea and real-world deal outcomes. During the collection of brain data, our participants were exposed to pitches without being prompted to evaluate them. These findings indicate that neural signals encode information that can predict subsequent behavior, even in the absence of explicit evaluation. This result persists even when controlling for individual characteristics, such as gender and race, as well as levels of dynamic capabilities. To demonstrate discriminant validity, we collected not only brain data but also other potentially predictive data, such as survey ratings and sentiment analysis of pitch transcripts, to evaluate whether the neural data provide additional predictive power. The brain measure emerged as the most robust predictor, while sentiment variables derived from pitch texts and survey measures, such as levels of *joy* sentiment, were not significant. Importantly, a one standard deviation increase in neural similarity among funders during the description of product solutions was associated with a roughly 1.4% higher likelihood of securing a deal, with the average deal amount climbing to approximately $364,000. Our findings indicate that electrical brain activity can serve as an additional predictive tool, complementing conventional stated measures and even fMRI measures (i.e., blood flow signals in the brain's reward system) [12,14,19].

Our neuroforecasting framework expands the theoretical boundaries of entrepreneurship research, particularly in the context of pitching. While previous studies in this area have relied upon post hoc survey methods [20], these approaches often face difficulties with accurate recall of past events [46] and are subject to limitations in accessing internal decision-making processes through surveys given to venture capitalists [21]. By contrast, utilizing continuous, real-time EEG data allows us to capture the audience's dynamic mental states and their immediate responses at specific moments during a pitch. We highlight the importance of the initial moments (thin-slice impressions) that capture investors' interest, as well as the subsequent moments formed by engaging storytelling that correlate with real-time deal outcomes—insights that traditional survey methods and functional MRI, which measures slowly-changing blood-flow signals (less than 0.5 Hz), are unable to provide. Both survey and fMRI measures are less effective at detecting moment-to-moment reactions during audiovisual pitch videos, making it difficult to pinpoint the specific timing that predicts successful outcomes. This is particularly important because studies show that the dynamics between entrepreneurs and investors can vary with the temporal structure of the pitch—not only in light of the order in which information is presented [47–49], but also in how this sequencing shapes communication effectiveness in pitching [34–37].

Surprisingly, we found that alpha asymmetry (FAA) measured within the first few seconds of each pitch significantly predicted investor interest in an independent, population-level sample. A founder speaking at a typical conversational speed in English of 120–150 words per minute would utter only 8–20 words in the first 4–8 seconds of a pitch. Thus our findings strongly support the idea that thin-slice impressions matter for gauging a broader audience and shaping downstream decision-making [30–32]. Beyond these early responses, cumulative FAA signals across the entire pitch duration also reliably predicted the amount of funding entrepreneurs secured in real outcomes. These findings highlight the importance for founders of creating an early positive impression in investors to secure a deal. These continuous neural measures explained variance in funding outcomes above and beyond traditional behavioral and textual predictors, reinforcing the utility of temporally sensitive physiological data in high-stakes contexts.

Notably, we also found that neural similarity across evaluators during the "product solution" segment—when the core value proposition of each venture was pitched—predicted funded from unfunded pitches. This segment functions as the climax of the pitch, and prior work on storytelling shows that audience similarity builds as a narrative unfolds and peaks at pivotal moments of meaning or resolution. Consistent with this pattern, neural similarity strengthened during this dense, high-stakes portion of the pitch, implying that effective storytelling aligns evaluators' neural responses in real time. Although individual FAA measures were not predictive here, neural similarity offered a distinctive signal of persuasive impact and deal success, in line with past work on neural similarity in audience forecasting [50] and communication effectiveness [17,34].

Our work also offers practical insights. For example, our study paradigm is scalable and can be applied in entrepreneurial competitions to predict which pitches will secure funding. Since the most important signals were localized to prefrontal cortex, portable EEG bands with sensors on the front of the head, rather than whole-head systems, may be sufficient for evaluating and reverse engineering pitches. Entrepreneurs who emphasize strong initial impressions and effective storytelling may foster greater neural similarity among potential funders during the presentation, which, in our study, was associated with increased investor interest and, in some cases, securing a deal. Additionally, entrepreneurs and trainers can draw on these neural insights to refine their presentation strategies and improve the likelihood of closing deals. Although we used recorded Shark Tank videos as stimuli, the same principles could inform how founders design and record pitches for crowdfunding platforms (e.g., Kickstarter). That said, our findings are specific to our experimental setting given that actual investment decisions often unfold over extended discussions and Q&A, which entail additional cognitive and emotional processes beyond those captured in our paradigm.

While this work is an initial step in neuroforecasting within entrepreneurship, its ecological validity and generalizability may be limited. EEG data were collected from a small laboratory focus group watching recorded 14 Shark Tank pitches rather than participating in live, interactive exchanges, and our behavioral benchmark came from an online convenience sample (MTurk) rather than professional investors. As a result, inferences may not generalize to broader investor populations—particularly experienced VCs or angel investors—whose expertise, incentives, and decision contexts (e.g., reciprocal attention, negotiation, or Q&A dynamics) differ from those of MTurk raters and lab viewers. As such, future work should expand to measuring brain signals in both investors and pitchers, in naturalistic settings incorporating structured Q&A, using hyperscanning EEG techniques to capture inter-brain synchrony during negotiation, as well as recruiting stratified samples that include professional investors to examine whether experience moderates the predictive value of EEG markers identified here.

## Acknowledgments

We acknowledge the participants at the *2024 Society for Neuroeconomics*, *2024 Academy of Management*, and *2025 Strategic Management Society* annual meetings for their helpful comments on earlier versions of this work.

## Author contributions

**Conceptualization:** Jin Ho Yun, Sohvi Heaton.

**Formal analysis:** Jin Ho Yun.

**Funding acquisition:** Michael L. Platt.

**Methodology:** Jin Ho Yun, JuiHsuan Sharon Wong.

**Project administration:** Michael L. Platt.

**Resources:** Michael L. Platt.

**Software:** Jin Ho Yun.

**Supervision:** Michael L. Platt.

**Writing – original draft:** Jin Ho Yun, Sohvi Heaton.

**Writing – review & editing:** JuiHsuan Sharon Wong, Peter Klein, Michael L. Platt.

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
