## [Decision Letter · Decision Letter 0]

1 Oct 2025

Dear Dr. YUN,

We look forward to receiving your revised manuscript.

Kind regards,

Claudia Noemi González Brambila, Ph.D.

Academic Editor

PLOS ONE

Journal Requirements:

4.Thank you for stating the following financial disclosure: [We thank R01MH095894, R01MH108627, R37MH109728, R21AG073958, R01MH118203,866 R56MH122819, and R01NS123054, the Wharton Behavioral Lab, and Wharton Neuroscience Initiative for funding the research.].

Reviewers' comments:

Reviewer's Responses to Questions

**Comments to the Author**

1. Is the manuscript technically sound, and do the data support the conclusions?

Reviewer #1: No

Reviewer #2: Partly

2. Has the statistical analysis been performed appropriately and rigorously?

Reviewer #1: No

Reviewer #2: Yes

3. Have the authors made all data underlying the findings in their manuscript fully available?

Reviewer #1: Yes

Reviewer #2: Yes

4. Is the manuscript presented in an intelligible fashion and written in standard English?

Reviewer #1: Yes

Reviewer #2: Yes

Reviewer #1: Review of PONE-D-25-41105: “Brain signals from new ventures predict entrepreneur fundraising success”

The authors evaluate the relationship between EEG data (from 28 participants with backgrounds in industry practices and/or business degrees who watched 14 Shark Tank pitch videos) and investment interest as measured from out-of-sample online survey data from the Amazon Turk (MTurk) Connect (N = 497) as well as actual Shark Tank pitching results: funded (1) or not funded (0).

The paper is well-organized and well-written. Exploratory studies such as the current one can be valuable additions to the literature, despite small samples sizes (low power), relatively large p-values, and lack of correction for multiple comparisons, even though they often prove irreplicable (Ioannidis 2005, “Why Most Published Research Findings Are False“, PLOS Medicine). However, fixed conclusions when interpreting such preliminary results would be premature.

Major issues:

1. Please use past tense when describing results, to avoid the implication that the results are globally applicable to all future studies that test the same hypotheses. Statements in need of reformulation include

Title: Could be more accurately stated as, for example, “Prediction of entrepreneur fundraising success from focus group EEG data,” although the difference is a subtle one.

Results: Sentences/phrases in need of modification include “Thus, greater FAA signals captured in early impressions to persuasive pitches can predict investment interest,” “… is the stronger predictor of aggregate fund amounts,” “These findings suggest that overall FAA signals can predict aggregate fund amounts raised,” and “showing that synchronized neural activity—particularly in the frontal-central regions, averaged across the entire pitch video—is predictive of both deal outcomes and funding amounts.”

Discussion: Sentences/phrases in need of modification include “As anticipated, our findings indicate that neural data from a laboratory sample predicts aggregate outcomes,” and “Entrepreneurs can focus on their initial impressions and the delivery of their product solutions to enhance neural synchrony among funders, which correlates with increased investor interest and ultimately securing a deal. In addition, entrepreneurs and trainers can use our neural insights to refine their presentation strategies and improve their chances of closing deals.”

2. Please be more specific in describing important details of the study in the title and abstract. For example, the “brain signals” discussed are EEG data. The “small laboratory sample” and “out-of-sample investment interest” could also benefit from additional details.

Minor items:

1. p4. VC is defined as venture capital, but VCs does not appear to be its plural form.

2. p11. Grammar: “data was.”

3. p14. Why the sudden switch to present tense in “We then derive the PSD reflecting the power at the alpha band frequency and apply a 5-second moving window across the EEG data”?

4. p15. The heading “Controls” might better be described as “Control variables” because the term controls usually refers to participants in a control group.

5. p20. Grammar: “However, when we employed our TFR rolling window technique and none of the time windows of moment-to-moment FAA activity forecasted binary deal outcomes.”

6. p25. Grammar: “While we represent an initial step in neuroforecasting within entrepreneurship, and several questions arise regarding our paradigm for future research.”

7. p25. “experienced vs. angel investors” seems to imply that these are mutually exclusive categories.

Reviewer #2: This manuscript presents an innovative study examining whether EEG signals from a small laboratory sample can predict investor interest in entrepreneurial pitches, as well as real-world funding outcomes. The work is timely and creative, bridging neuroforecasting with entrepreneurship research. The combination of lab EEG, out-of-sample behavioral data, and actual fundraising results is a major strength. The manuscript is clearly written and the motivation well framed.

Please find my comments below:

1. Sample size and generalizability

• The lab EEG sample (N=28) is small for inter-subject correlation and synchrony analyses. While the out-of-sample survey (N=497) helps, the EEG-derived predictions might be unstable or overly sensitive to outliers.

• Suggestion: report power analysis or bootstrapping stability checks; highlight limits in generalization to broader investor populations (e.g., experienced VCs vs. MTurk participants).

2. Ecological validity

• Pitches are Shark Tank videos, not live, interactive pitches. EEG signals may differ from actual investor-founder interactions that include Q&A and dynamic feedback.

• Suggestion: acknowledge stronger ecological limitations and recommend testing in dyadic or live pitch settings.

3. EEG preprocessing and robustness

• Preprocessing is described, but critical details (e.g., ICA rejection thresholds, % variance removed, number of components rejected per subject) are not fully reported.

• FAA computation relies on F3/F4; however, alpha asymmetry is notoriously noisy and can be confounded by artifacts (eye movement, muscle tension).

• Suggestion: add robustness checks (e.g., alternative reference schemes, different band definitions, inclusion/exclusion of noisy subjects).

4. Multiple comparisons / statistical control

• Rolling-window FAA analyses involve many time bins, but correction for multiple comparisons is not clearly described (e.g., cluster-based permutation, FDR).

• Synchrony effects are reported around 47–50 seconds — was this result corrected for the fact that many windows were tested?

• Suggestion: clarify family-wise error control to strengthen claims.

5. Interpretation risk (“reverse inference”)

• FAA is interpreted as “approach motivation” and synchrony as “collective engagement.” These are plausible, but EEG signals are indirect markers and may reflect multiple overlapping processes (e.g., visual attention, motor planning).

• Suggestion: soften causal language; frame findings as predictive markers rather than mechanistic explanations.

6. Behavioral benchmarks

• Sentiment analysis was limited (VADER, NRCLex) and may not capture nuance (e.g., sarcasm, authenticity). The null findings could reflect weak measures rather than EEG superiority.

• Suggestion: consider adding linguistic richness measures (e.g., LIWC categories, narrative complexity) or justify why the chosen sentiment models are adequate.

Overall, the study is technically sound and provides novel insights into the neural correlates of entrepreneurial success. However, the statistical analyses need more transparency and caution. The main conclusions are supported, but they should be framed more conservatively, especially regarding real-world outcome prediction and the robustness of temporal effects.

**Do you want your identity to be public for this peer review?** For information about this choice, including consent withdrawal, please see our Privacy Policy

Reviewer #1: No

Reviewer #2: No

---

## [Author Response · Author response to Decision Letter 1]

27 Nov 2025

Thank you so much for offering a revision opportunity. Hereby we are submitting the revision work.

---

## [Decision Letter · Decision Letter 1]

23 Dec 2025

Predicting Entrepreneur Fundraising Success from Focus Group EEG Data

PONE-D-25-41105R1

Dear Dr. YUN,

We’re pleased to inform you that your manuscript has been judged scientifically suitable for publication and will be formally accepted for publication once it meets all outstanding technical requirements.

Kind regards,

Claudia Noemi González Brambila, Ph.D.

Academic Editor

PLOS One

Additional Editor Comments (optional):

Reviewers' comments:

Reviewer's Responses to Questions

**Comments to the Author**

Reviewer #1: All comments have been addressed

Reviewer #2: All comments have been addressed

2. Is the manuscript technically sound, and do the data support the conclusions?

Reviewer #1: (No Response)

Reviewer #2: (No Response)

3. Has the statistical analysis been performed appropriately and rigorously?

Reviewer #1: (No Response)

Reviewer #2: (No Response)

4. Have the authors made all data underlying the findings in their manuscript fully available?

Reviewer #1: (No Response)

Reviewer #2: (No Response)

5. Is the manuscript presented in an intelligible fashion and written in standard English?

Reviewer #1: (No Response)

Reviewer #2: (No Response)

Reviewer #1: (No Response)

Reviewer #2: (No Response)

**Do you want your identity to be public for this peer review?** For information about this choice, including consent withdrawal, please see our Privacy Policy

Reviewer #1: No

Reviewer #2: No

---

## [Editor Report · Acceptance letter]

PONE-D-25-41105R1

PLOS One

Dear Dr. YUN,

I'm pleased to inform you that your manuscript has been deemed suitable for publication in PLOS One. Congratulations! Your manuscript is now being handed over to our production team.

Kind regards,

on behalf of

Dr. Claudia Noemi González Brambila

Academic Editor

PLOS One